# Impact of Agricultural Extension Services on Fertilizer Use and Farmers' Welfare: Evidence from Bangladesh

Mohammad Mahbubur Rahman *[image: ORCID] and Jeffry D. Connor

Centre for Markets, Values and Inclusion, UniSA Business, University of South Australia, Adelaide, SA 5001, Australia; jeff.connor@unisa.edu.au
* Correspondence: rahmm009@mymail.unisa.edu.au

**Abstract:** Although many studies have assessed the impact of extension, most treat the presence or absence of extension as a binary variable to test treatment effects, and fewer investigate how the type of provider (e.g., govt./private) and the frequency of the contact (number of extension visits) impact farm household welfare. To address this knowledge gap, this article investigates the impact of agricultural extension access, frequency, and provider type on chemical fertilizer application, crop yield, and profit. Data from a nationwide survey in 2015 in Bangladesh, a case country with a heavy over-application of urea fertilizer, are the basis for the endogenous switching regression approach to control for potential self-selection and endogeneity. The empirical results revealed significant differences in the outcomes for farmers who had just one extension contact, more than one extension contact, and those who accessed private provisions. We found that farmers who frequently accessed extension used significantly less urea fertilizer than farmers who accessed extension only once. Farmers who accessed extension more frequently also experienced a statistically significantly higher yield and profit from cropping. Private extension access appeared to result in statistically significantly higher incomes but not reduced urea fertilizer application rates. Our results suggest that a more nuanced understanding can be gained from extension source and frequency treatment effects modelling than with the presence or absence of the extension binary variable formulation that is most common in the literature.

**Keywords:** fertilizer use; yield; net income; government extension; private extension; profit

## 1. Introduction

In many parts of the world, increasing agricultural production is an important strategy to increase income, reduce hunger, and improve other measures of well-being [1]. Few countries have achieved sustained economic growth without first or simultaneously developing their agricultural sector [2]. Agricultural extension is a mechanism by which new technologies, more effective management options, and better farming practices can be transmitted to farmers [2]. Extension can reduce productivity differentials amongst farmers by accelerating technology transfer, increasing farmers' knowledge, and assisting them in improving their farm management practices [3]. Thus, agricultural extension services provide human-capital-enhancing inputs and information that can potentially improve rural welfare [4,5].

The value of agricultural extension services in increasing value, productivity, and food security is widely recognized [6–8]. Most previous studies evaluating agriculture extension show a positive impact on farm productivity [6,8–10], farm technical efficiency [10–12], net farm income [6,13,14], technology adoption [15], and poverty reduction [6,7].

One limitation in the literature is that most studies only evaluate extension contact presence or absence as a binary explanatory variable. This ignores the impacts of the quality and diversity of extension, despite good reasons to assume that the effects of the extension

are likely to vary depending on how the service is delivered and the recipient's circumstances [11,16]. There are a few exceptional studies that do account for the attributes of extension. For example, Hasan and Otsuki (2011) [10] showed that total factor productivity was approximately five times higher among farmers participating in private extension programs than for farmers participating in government programs. Another example is Ragasa and Mazunda (2018) [16], who found that the receipt of extension advice had a consistently insignificant effect on crop production and food security, except for farmers who found the extension to be 'very useful'.

Although most of the literature on the impact of extension services focuses on yield and net farm profit, this article focuses on how extension advice influences farmers' overuse of chemical fertilizers. A good amount of literature demonstrates the excessive use of chemical fertilizer in developing countries, which can be up to five times the recommended dose [17–20]. Although there are studies that investigate the impact of extension on chemical fertilizer adoption [15], few studies evaluate the impact of extension on chemical fertilizer application levels.

This paper contributes to the literature by evaluating how fertilizer use, yield, and net profit are influenced by the various attributes of the extension services that farmers receive including the frequency of extension contacts (one or more than one contact in a year) and the mode of delivery (from government or private sources). Hence, our hypothesis to be tested in the context of the overuse of fertilizer in Bangladesh is that extension access should result in lower rates of nitrogen fertilizer application that will reduce the excess burden of the cost of production and lead to higher net farm profit.

A challenge with evaluating the impact of extension is that farmers who self-select extension access and the non-access groups also often systematically differ in observed and unobserved characteristics such as their abilities, desires, risk preferences, and aspirations. Such treatment and control group differences can potentially influence the outcomes of interest and be alternative explanations for the treatment of the outcome differences between the treatment and control groups. The unbiased estimation of the impact of extension should account for self-selection bias [21]. This paper uses an endogenous switching regression approach to robustly address endogeneity and self-selection.

Section 2 of the paper discusses the extent of the agricultural extension service provisions in Bangladesh, the types of providers, how the extension has focused on providing advice on fertilizer use, and what is known about fertilizer overuse and the effectiveness of the extension in addressing the issue. Section 3 describes the data and provides a summary of the statistics, Section 4 describes the study's methodology, Section 5 presents the results, and finally, in Section 6, we offer the discussion and conclusion.

## 2. Evidence of Impact of Extension on Development Outcomes and Fertilizer Application

Worldwide, there is evidence of the overuse of chemical fertilizer. In China, the overuse of chemical fertilizer is well documented [22–25]. Zikria and Damayanti (2019) [26] reported the overuse of chemical fertilizer in Indonesia and Beshir et al. (2012) [17] reported the excessive use of chemical fertilizer in Ethiopia. Islam and Hossain (2021) [19] found that farmers in Bangladesh apply two to five times the recommended chemical fertilizer application rates. Many factors are identified as being related to the overuse of chemical fertilizer, such as land size [27], the risk aversion behavior of farmers [23], a lack of knowledge regarding fertilizer use, the absence of agricultural extension services, and misleading information provided by fertilizer retailers [28,29]. Fertilizer overuse has multiple effects: it increases production costs and, as a result, reduces net farm income [30], causes economic losses [27], reduces fertilizer utilization rates for crops [31], reduces soil micronutrients [32], reduces soil fertility [19], causes non-point source pollution [33] and greenhouse gas emissions [34], and results in harm to the environment, water, soil, atmosphere, biology, and human health [27]. It follows that reducing the overuse of chemical fertilizers can improve

the efficiency, cost-effectiveness, and resilience of agricultural systems and improve farmer and environmental outcomes [35].

The focus here is on a comprehensive and reliable evaluation of the impact of agricultural extension on reducing chemical fertilizer overuse. This is important considering the positive impact of agricultural extension on many other outcomes and because governments spend millions of dollars to support extension activities [36–39].

Extension agents advise farmers on various aspects of crop production and management including optimal fertilizer use [40]. Besides the transfer of technology and knowledge, extension services identify and document farm innovations, assist in the distribution of fertilizer and other agricultural inputs, and facilitate rural programs dictated by national policies [41]. The central hypothesis investigated here is that farmers who come in contact with extension agents will apply fertilizer at lower rates (closer to the recommended rates) than farmers who do not come in contact with extension agents. Because they save the expense of overapplication, they will also experience higher net profit. We further hypothesize that fertilizer application rates, yield, and net profit will differ for farms having more frequent extension contact and for farms receiving private as opposed to public extension advice.

To check these hypotheses, Bangladesh is chosen as the country for this case study. Over 47% of the country's total labor force is engaged in agriculture and 70% of the population lives in rural areas with their livelihood and wellbeing dependent on agriculture [42]. Further, there is evidence of chemical fertilizer overuse in Bangladesh [19].

Bangladesh has a long history of extension activities since 1906 when the first agricultural department was established there [43]. Bangladesh now has an extension service network spread across the country in every district [44]. Despite their widespread presence and multifaceted contribution to farmers' knowledge, empirical evidence on the relevance, efficacy, and effectiveness of Bangladesh's agricultural extension services is lacking. The scarcity of good quality local studies motivated this evaluation of the impact of various forms of extension services and delivery modes on farmers' welfare in Bangladesh.

## 3. Methodology

There are several ways to address the potential sources of bias in this study (endogeneity and self-selection) ranging from instrumental variable methods to endogenous switching regression to experimental and quasi-experimental methods [45–48]. Our study deploys the endogenous switching regression (ESR) method, a well-recognized approach to treating endogeneity, including self-selection, which is likely in our study [48,49]. The ESR method is similar to the Heckman two-step approach and is especially useful when the treatment is not randomly distributed among the treatment and control groups [50]. This method is popular for exploring food production issues and considering farmers' behavior [51–55]. The ESR approach addresses endogeneity by estimating the selection and outcome equations simultaneously using a full information maximum likelihood approach [48,49,56,57].

In the ESR model, we specify the selection equation for a recipient of an extension service as:

$$T_i^* = X_i\alpha + \delta_i \text{ with } T_i = \begin{cases} 1 \text{ if } T_i^* > 0 \\ 0 \text{ otherwise} \end{cases} \tag{1}$$

That is, a farmer will opt to access an extension service ($T_i = 1$), if $Y^* > 0$, where $Y^*$ represents the expected benefits of accessing the extension service.

The relationship between a vector of explanatory variables $X$ that determines a farmer's propensity to access an extension service and the outcome $Y$ conditional on treatment can be represented as follows:

$$Y_{1i} = X_{1i}\beta_1 + \varepsilon_{1i} \text{ if } T_i = 1 \tag{2}$$

$$Y_{2i} = X_{2i}\beta_2 + \varepsilon_{2i} \text{ if } T_i = 0 \tag{3}$$

The outcomes of interest in this study—$Y_i$ is the yield (ton/ha), net farm profit (Taka per hectare, or Tk/ha), and nitrogen fertilizer application rate (kg/ha) and $X_i$ represents a vector of the explanatory variables. $\varepsilon_i$ is the regression error term, assumed to have a trivariate normal distribution, with a zero mean and covariance matrix. The estimated covariance between $\delta$ and $\varepsilon$'s ($\rho_1$ and $\rho_2$, respectively) is the transformation of the correlation between the errors in the "switching regression model with endogenous switching" [58].

Identification of the ESR model requires at least one additional variable as an instrument. The selection of instrumental variables should directly affect the selection variable but not the outcome variable. In this study, we used the distance from the bazar (village market) and the use of mobile phones as the selection instrumental variables and checked the admissibility of the instruments by performing a simple falsification test: if a variable is a valid selection instrument, it will affect the households of farmers who receive extension services but will not affect the outcome variables of the households of farmers who do not receive extension services (Appendix B).

In addition to using the endogenous switching regression model, we calculated the farms' actual observed outcome for fertilizer use, yield, and net farm profit in the two cases presented below and the difference as the observed treatment effect:

$$E(Y_{1i}|T_i = 1) = \left[\sum_{Ti=1}(X_{1i}\beta_1 + \sigma_{1n}\gamma_{1i})\right]/N_1 \tag{4}$$

$$E(Y_{2i}|T_i = 0) = \left[\sum_{Ti=0}(X_{2i}\beta_2 + \sigma_{2n}\gamma_{2i})\right]/N_0 \tag{5}$$

$N_1$ and $N_0$ are the number of observations with $Ti = 1$ and $Ti = 0$, respectively.

## 4. Data and Summary of Statistics

### 4.1. Data

This study used data from the Bangladesh Integrated Household Survey (BIHS, 2015) administered by IFPRI (International Food Policy Research Institute), which is a nationally representative survey of 5500 households that covers all administrative divisions in Bangladesh. The BIHS questionnaire comprises several modules with questions related to living standards including the level of education, employment, household demographics, assets and their values, profit, expenditure, microcredit, disaster mitigation, and migration ad remittance.

The primary focus of the survey was to collect data on agricultural production, input use, costs, and return at the plot level. This includes data on crop farming activities, yields, input uses for seed, fertilizers, pesticides, the technology used, land characteristics, land management, extension service access, and other issues. The survey questionnaire had a separate section regarding extension services, which included questions about (a) whether the farmer had contact with extension agents or not, (b) whether or not the extension advice was related to fertilizers, (c) how many times the extension agent contacted farmers, (d) whether farmers were satisfied with the service they received, (e) in which mode the service was delivered (face-to-face or by mobile phone), and (f) whether the service they accessed was from a government or private service provider. For the treatment variables, we used (a) the presence or absence of an extension agent as a binary variable, (b) one contact and more than one extension contact as a binary variable, (c) receipt of extension advice from a government extension agent is a binary variable, and (d) receipt of extension services from a private provider as a binary variable. As an outcome variable, we used per hectare nitrogen fertilizer (urea) applied, per hectare yield, and per hectare net profit. We focused on a single crop, aman rice, to avoid challenges related to different fertilizer requirements for different crops.

The control variables we used in this study should demonstrate a considerable impact on crop production (Table 1). These variables have also been used in previous studies. For

example, Elias et al. (2013) [9] used the household head's age, gender, education, land size, livestock, use of credit, plot distance from home, and measures of various production inputs, Wossen et al. (2017) [48] used the household head's age, gender, education, marital status, land size, mobile phone access, use of credit, plot distance from agricultural input dealers, and production inputs; Ragasa and Mazunda (2018) [16] used the household head's age, gender, education, land size, child-dependency ratio, household size, annual rainfall, use of credit, plot distance from the nearest market, and an asset index.

**Table 1.** Summary statistics of characteristics of farmers receiving different types of extension services.

| Item | Ext. Service Receiver vs. No Ext. | | | More than One Ext. Contact vs. No Ext. | | Govt. Ext. Service Receiver vs. No Ext. | | Private Ext. Service Receiver vs. No Ext. | |
|---|---|---|---|---|---|---|---|---|---|
| | Ext. | No-ext | *t*-Test | More One | *t*-Test | Govt. | *t*-Test | Receiver | *t*-Test |
| | Mean | Mean | | Mean | | Mean | | Mean | |
| Yield/ha (MT) | 3.40 | 3.22 | −0.17 *** | 3.44 | −0.21 *** | 3.3 | −0.07 | 3.58 | −0.36 ** |
| Per ha net profit (BDT) | 28,585 | 27,753 | −832 | 29,770 | −2028 * | 28,417 | −666 | 28,294 | −540 |
| Per ha Urea use (kg) | 116.32 | 116.53 | 0.21 | 113 | 3 | 110 | 6.02 | 126 | −10 |
| Land area (decimal) | 29.24 | 27.82 | −1.42 * | 29.47 | −1.6 * | 29.23 | −1.3 | 32.25 | −4.4 * |
| Household head's age (years) | 49.33 | 47.46 | −1.87 *** | 49.3 | −1.8 *** | 49.58 | −2.12 *** | 46.77 | 0.69 |
| Education of the farmer | 5.14 | 3.57 | −1.57 *** | 5.5 | −2 *** | 5.04 | −1.46 *** | 4.29 | −0.71 * |
| Household head's gender | 0.99 | 0.94 | −0.04 *** | 0.99 | −0.04 *** | 0.99 | −0.04 *** | 1 | −0.05 *** |
| Agriculture as the main occupation | 0.33 | 0.28 | −0.04 *** | 0.33 | −0.5 *** | 0.38 | −0.09 *** | 0.13 | 0.14 *** |
| Own irrigation | 0.14 | 0.08 | −0.06 *** | 0.16 | −0.07 *** | 0.15 | −0.07 *** | 0.10 | 0.02 |
| Labor hours (per hectare) | 585 | 563 | −22 | 576 | −13 | 547 | 19 | 765 | 203 *** |
| Having livestock | 0.92 | 0.92 | 0.00 | 0.92 | 0.01 | 0.91 | 0.01 | 0.93 | 0.00 |
| Improved variety | 0.89 | 0.88 | −0.006 | 0.93 | −0.05 ** | 0.83 | 0.05 ** | 0.67 | 0.21 *** |
| Access to credit | 0.80 | 0.76 | −0.03 *** | 0.8 | −0.03 ** | 0.8 | −0.03 *** | 0.85 | −0.08 ** |
| Mobile phone | 0.99 | 0.97 | 0.02 *** | 0.98 | 0.017 ** | 0.99 | −0.01 ** | 0.98 | 0.01 |
| Distance to market (km) | 0.60 | 0.61 | 0.009 | 0.58 | 0.03 | 0.61 | 0.00 | 0.53 | 0.07 |
| No of observations | 1209 | 5794 | | 925 | | 898 | | 164 | |

* Significant at the 10% level,** Significant at the 5% level, *** Significant at the 1% level. The first three columns show the means and a *t*-test comparison between farmers who did and did not receive extension services. In the following columns, as the control group (farmers who did not receive extension services) is the same, only the treatment group means and treatment and the control *t*-test comparisons are reported.

Some farmers received both government and private services. In this case, to test the impact of government versus private services, we dropped the 128 farmers who received both private and government services.

### 4.2. Summary of Statistics

The expected values and t-statistics for the differences between the treatment and control group values for the variables included in the endogenous switching regression (ESR) method are presented in Table 1. The table also presents the expected values and t-tests for the differences between the sub-groups, including those receiving one and more than one extension visit, government extension services, and private extension services, in comparison with the farmers who received no extension services. In every case, we compare the sub-group receiving a particular type of extension with the farmers who did not receive any extension services.

The statistics summary shows that around 16 percent of survey participants received agriculture extension services and that farmers who received extension services had higher yields, more ownership of irrigation pumps, greater access to credit, and larger fields on average and were older, more educated, more likely to have agriculture as their main

occupation, and more likely to be male. Farms where extension services had visited experienced less rainfall.

The comparison between the farmers who had more than one extension contact and those who had no extension contacts is very similar to the first group comparison except for the significantly higher net farm income for the farmers that had more than one extension contact. The farmers who had more than one extension contact had more school years, more ownership of irrigation pumps, and a higher net farm income.

The comparison between the farmers who received government extension services and those who received no extension services shows that farmers who received government extension services had more school years, older household heads, more frequently chose agriculture as their main occupation, and had more irrigation pumps. There was no significant difference in the plot area, per hectare labor use, total urea fertilizer use, phosphorous fertilizer use, yield, and net income.

On the other hand, the subgroup comparison between those who received private extension services and those who received no extension services tells us that the farmers who received private extension services had larger land holdings, used much more phosphorous fertilizer, had higher yields, spent less time in the field, and less agriculture as their main occupation. However, there were no significant differences in the ownership of irrigation pumps, household head age, distance to market, and urea fertilizer use.

## 5. Results

The endogenous switching regression model results for the impact of the various forms of extension services on urea utilization are presented in Table 2 below.

The first part of the endogenous switching regression results is for the outcome equation (per hectare urea application rate) conditional on receiving treatment (extension) ($Ti = 1$) and includes the household head's age, whether agriculture is their main occupation, access to credit, labor hours, access to irrigation, and the use of an improved variety of seeds. The second part of the results reports the urea application rate conditional on non-participation in extension services ($Ti = 0$). Household head's age, education, rice area, and own irrigation are all found to be negatively associated with the urea fertilizer use for this control group.

The third part of Table 2 is the results of the selection equations that estimate the propensity to participate in extension services. We observe that the variables, household head's age, gender, education, own irrigation, and access to credit, are positively associated with extension participation. The ESR regression coefficients for the yield and profit equations are presented in Appendix A. Our interest is in the average treatment effects. Hence, we calculate the observed values of the dependent variables (urea application, yield, and net farm profit) on farms that received extension services and on farms that did not receive the treatment. The observed value of the difference is the observed treatment effect of receiving extension services.

### 5.1. Fertilizer Use

The analysis shows that having extension contact reduced farmers' nitrogen fertilizer use. However, the impact is greater for the farmers who had more than one extension contact. Farmers that received government extension services used similar amounts of nitrogen fertilizer compared to those that did not receive extension services. Similarly, farmers receiving private extension support used a similar amount of nitrogen fertilizer compared to the farmers that did not receive extension services (see Table 3).

**Table 2.** Impact of extension service on urea fertilizer use (ESR).

| Variables | Ext. Receiver vs. No Ext. | More than One vs. No Ext. | Govt. ext. vs. No Ext. | Private ext. vs. No Ext. |
|---|---|---|---|---|
| Per ha Urea Use (Participation = 1) | | | | |
| Household head's age | 1.694 *** | 1.560 *** | 1.955 *** | 1.756 *** |
| Household head's gender | −112.2 | −205.4 *** | | |
| Household head's education | −1.681 | −1.757 | −0.558 | 2.155 |
| Agriculture as the main occupation | 44.27 * | 50.22 ** | 27.64 | 80.94 * |
| Rice area | −0.720 *** | −0.683 *** | −0.789 *** | −0.803 *** |
| Access to credit | 21.95 ** | 37.59 *** | 13.74 | 89.29 *** |
| Own irrigation | 21.84 * | 36.99 *** | 8.033 | |
| Labor hours | 0.0223 *** | 0.0267 *** | 0.0245 ** | 0.0387 *** |
| Improved variety | 52.66 *** | 44.41 *** | 66.85 *** | |
| Having livestock | −19.83 | −22.20 * | −4.588 | −28.8 |
| Constant | −15 | −12.96 | −17.96 | −37.61 |
| Per ha Urea use (Participation = 0) | | | | |
| Household head's age | −0.953 *** | −0.900 *** | −0.864 *** | −0.403 *** |
| Household head's gender | 1.847 | 9.61 | | |
| Household head's education | −3.011 *** | −3.369 *** | −2.542 *** | 0.314 |
| Agriculture as the main occupation | 2.883 | 0.5 | 1.448 | 5.884 |
| Rice area | −0.596 *** | −0.584 *** | −0.565 *** | −0.561 *** |
| Access to credit | −0.0626 | −0.0612 | −0.0603 | −0.0535 |
| Own irrigation | −15.91 *** | −15.39 *** | −11.55 *** | −7.180 ** |
| Labor hours | −3.903 | −3.826 | −3.787 | −3.336 |
| Improved variety | 5.412 | 5.262 | 1.575 | |
| Having livestock | −5.812 | −5.691 | −5.614 | |
| Constant | 147.0 *** | 146.0 *** | 137.0 *** | 165.2 *** |
| Participation in extension services | | | | |
| Household head's age | 0.00595 *** | 0.00578 *** | 0.00591 *** | 0.00286 |
| Household head's gender | 0.431 *** | 0.371 ** | | |
| Household head's education | 0.0303 *** | 0.0401 *** | 0.0239 *** | 0.0474 *** |
| Agriculture as the main occupation | −0.188 ** | −0.164 * | −0.139 | −0.228 |
| Rice area | 0.00309 *** | 0.00324 *** | 0.00339 *** | 0.00042 |
| Access to credit | 0.172 *** | 0.200 *** | 0.121 *** | 0.265 *** |
| Own irrigation | 0.191 *** | 0.213 *** | 0.226 *** | |
| Labor hours | −0.000132 *** | −0.000167 *** | −0.000190 *** | 0.000136 *** |
| Improved variety | −0.0867 *** | −0.0521 * | −0.173 *** | |
| Having livestock | 0.00994 | 0.0067 | −0.0179 | 0.218 * |
| Mobile phone | 0.0750 *** | 0.0561 *** | 0.0603 *** | 0.0705 * |
| Distance to market | 0.144 | 0.1 | 0.235 ** | |
| Constant | −1.881 *** | −1.958 *** | −1.509 *** | −2.652 *** |
| Rho 1 | −0.01 | −0.04 | −0.07 | 1.11 *** |
| Rho 2 | −2.1 *** | −2.1 *** | −2.3 *** | 0.02 |
| Observations | 6564 | 6315 | 6440 | 6564 |
| Prob > chi2 | 0.00 | 0.00 | 0.00 | 0.00 |

* Significant at the 10% level, ** Significant at the 5% level, *** Significant at the 1% level.

**Table 3.** Treatment effect of various forms of extension services on fertilizer use.

| Name of Fertilizer | Fertilizer Use (ESR)(Observed Treatment Effect) | | | | | | | |
| --- | --- | --- | --- | --- | --- | --- | --- | --- |
| | Extension Contact | | More than One Contact | | Govt. Ext. Service | | Private Ext. Service | |
| | $E(Y_{1i}|T_i=1)$ | $E(Y_{2i}|T_i=0)$ | $E(Y_{1i}|T_i=1)$ | $E(Y_{2i}|T_i=0)$ | $E(Y_{1i}|T_i=1)$ | $E(Y_{2i}|T_i=0)$ | $E(Y_{1i}|T_i=1)$ | $E(Y_{2i}|T_i=0)$ |
| Per hectare Urea | 176.71 | 180.81 | 174.37 | 180.81 | 179.16 | 179.94 | 172.36 | 172.32 |
| | =−4.09 *** | | =−6.44 *** | | =−0.78 | | =0.037 | |

*** Significant at the 1% level.

### 5.2. Yield and Profit

Although having extension contacts (described with a binary indicator) had a statistically significant treatment effect on yield (Table 4), this higher yield did not convert to a statistically significant net farm profit effect (Table 5). This may be because these farms used more input and as a result, their production costs were higher. On the other hand, there were positive and significant yield (Table 4) and net profit (Table 5) treatment effects for farms that had more than one extension contact. The yield treatment effects were not significant for farms that received government extension services. A positive and significant profit effect was observed for the farms that received private extension services.

**Table 4.** Impact of various forms of extension on yield.

| | ESR (Observed Treatment Effect) | | | | | |
| --- | --- | --- | --- | --- | --- | --- |
| | Extension Contact | | More than One Contact | | Govt. Ext. Service | |
| | $E(Y_{1i}|T_i=1)$ | $E(Y_{2i}|T_i=0)$ | $E(Y_{1i}|T_i=1)$ | $E(Y_{2i}|T_i=0)$ | $E(Y_{1i}|T_i=1)$ | $E(Y_{2i}|T_i=0)$ |
| Per hectare yield | 3.65 | 3.47 | 3.75 | 3.47 | 3.49 | 3.48 |
| | =0.18 *** | | =0.28 *** | | =0.01 | |

*** Significant at the 1% level.

**Table 5.** Impact of various forms of extension on net farm profit.

| | ESR (Observed Treatment Effect) | | | | | | | |
| --- | --- | --- | --- | --- | --- | --- | --- | --- |
| | Extension Contacts | | More than One Contact | | Govt. Ext. Service | | Private Ext. Service | |
| | $E(Y_{1i}|M_i=1)$ | $E(Y_{2i}|M_i=0)$ | $E(Y_{1i}|M_i=1)$ | $E(Y_{2i}|M_i=0)$ | $E(Y_{1i}|M_i=1)$ | $E(Y_{2i}|M_i=0)$ | $E(Y_{1i}|M_i=1)$ | $E(Y_{2i}|M_i=0)$ |
| Net farm profit | 20,951 | 21,601 | 22,980 | 21,596 | 18,885 | 21,729 | 26,330 | 21,229 |
| | =−650 ** | | =1383 *** | | =−2843 *** | | =5101 *** | |

,** Significant at the 5% level, *** Significant at the 1% level.

Yield outcome of private extension services cannot be determined due to concavity issues as the sample size is very small and the variation in the output is not differentiated enough.

### 5.3. Robustness Check with PSM

We also tested all hypotheses using a PSM method to check the robustness of our conclusions about the impact of extension on fertilizer use, yield, and profit. The PSM results reported in Appendix C allow mostly similar conclusions to those based on the ESR method.

## 6. Discussion and Conclusions

### 6.1. Summary

This paper aimed to address the absence of the evaluation of the impact of the frequency of extension contact and provider type on fertilizer input use, yield, and profit metrics in contrast to the predominant practice of treating extension contacts as a binary

variable. As there is evidence of the overuse of chemical fertilizer in our case study country and public extension services in the country target the issue, our key hypotheses were that having an extension contact reduces chemical fertilizer application rates, which ultimately reduces production costs and increases net farm profit. We also tested the hypotheses that private versus public and more versus less extension access impact fertilizer application, yields, and profits differently. Because extension participation is voluntary, there is a possibility of self-selection and endogeneity. To overcome the bias in the estimations that may arise, we employed endogeneity switching regression.

The results revealed that the traditional binary variable model provided a significantly less nuanced perspective on the impact of extension than we gained from estimating the impact of extension frequency and the sources of the provisions' treatment variables. The binary indicator model found that extension had a significant negative impact on nitrogen fertilizer use. The impact on fertilizer application reduction was even greater for the farmers who had multiple extension contacts. Although private extension services were estimated to have the greatest profit benefits, they did not reduce the application rate of urea fertilizer. More frequent extension contact resulted in both more profit and reduced fertilizer application. Hence, we suggest, that more intensive extension contact is the better approach to reducing the overuse of urea fertilizer, simultaneously gaining more yield and profit.

*6.2. Discussion*

There are significant critics of extension services as a provision for the 'public good'. It has been described as ineffective in many developing countries [59] and a persistent failure [60]. Many authors suggest that extension services should be primarily offered by private providers [14,61]. Our study supports the notion that private provisions may provide greater net income benefits to farmers than public provisions. However, we conclude, that a more intensive extension service (public or private) may be the better option for reducing the overuse of chemical fertilizer while simultaneously increasing yield and profit.

Our findings that extension contact measured without accounting for the frequency or source of extension did not impact farm profitability contrasts with some related past studies that did find that extension contact enhanced farm productivity and profit (e.g., [6]). The findings from our regressions based on the more nuanced description of extension support other similar results such as the conclusion by Lyne, Jonas, and Ortmann (2018) [14] that outsourced or private extension services enhanced farm productivity and profit.

Though many studies assess the impact of extension on crop yield and profit, fewer deal with the impact of agricultural extension on the overutilization of chemical fertilizers. The findings of this paper are relevant not only in developing countries but also in developed countries where the overapplication of fertilizer is reported [19,27,30,62].

One of our study's limitations is that the data set we used does not include variables measuring land quality or soil type differences. It is an important missing variable because precise fertilizer recommendations vary with these attributes. Another limitation is that we only estimated the impact of extension on one type of fertilizer, nitrogen. Other fertilizer types (P and K) can be investigated in further studies. Another limitation is the BIHS 2015 data used for this study as it is inevitable that some of the variable values driving the regression results have changed and are now different than they were in 2015. This also implies that a similar regression with more recent survey data, if available, would produce somewhat different results. However, given that there were no major natural disasters or policy changes after 2015 that impacted the agricultural sector and that many key outcome determinants are slow-moving, such as levels of education, we suspect that the key results would not be too different for newer survey data. As there were no major natural disasters or policy changes after 2015 impacting the agricultural sector, this dataset can be treated as valid till now.

**Author Contributions:** Conceptualization, M.M.R. and J.D.C.; Data curation, M.M.R.; Formal analysis, M.M.R.; Investigation, M.M.R.; Methodology, M.M.R.; Software, M.M.R.; Supervision, J.D.C.; Writing—original draft, M.M.R.; Writing—review & editing, J.D.C. All authors have read and agreed to the published version of the manuscript.

**Funding:** This research received no external funding.

**Data Availability Statement:** In this study we have used Bangladesh Integrated Household Survey (BIHS, 2015) data which was administered by IFPRI (International Food Policy Research Institute). This dataset is readily available for public in Harvard Dataverse which is available at the following link: https://dataverse.harvard.edu/dataset.xhtml?persistentId=doi:10.7910/DVN/BXSYEL.

**Conflicts of Interest:** The authors declare no conflict of interest.

## Appendix A

**Table A1.** ESR (Impact on Yield).

| Variables | Ext. Receiver vs. No Ext. | More than One vs. No Ext. | Govt. Ext. vs. No Ext. |
|---|---|---|---|
| Per ha Yield (Participation = 1) | | | |
| Household head's age | −0.00354 | −0.00705 | −0.00545 |
| Household head's gender | −3.150 *** | −6.774 *** | −3.645 *** |
| Household head's education | 0.0184 | −0.112 *** | −0.00102 |
| Agriculture as the main occupation | 0.268 | 0.848 ** | 0.211 |
| Rice area | −0.00334 * | −0.00682 *** | −0.00107 |
| Access to credit | 0.250 * | −0.288 * | 0.297 ** |
| Own irrigation | 0.347 ** | −0.328 * | |
| Labor hours | 0.000437 *** | 0.000214 * | 0.000516 *** |
| Improved variety | 0.870 *** | 1.040 *** | 1.091 *** |
| Having livestock | −0.22 | −0.153 | −0.254 |
| Constant | 4.597 *** | 13.29 *** | 5.221 *** |
| Per ha Yield (Participation = 0) | | 37.89 | −244.7 |
| Household head's age | −0.00868 *** | −0.00912 *** | −0.00857 *** |
| Household head's gender | 0.816 *** | 0.736 *** | 0.792 *** |
| Household head's education | 0.0514 *** | 0.0519 *** | 0.0482 *** |
| Agriculture as the main occupation | 0.147 | 0.190 * | 0.152 |
| Rice area | −0.00149 * | −0.00160 ** | −0.00158 ** |
| Access to credit | −0.117 ** | −0.129 *** | −0.121 ** |
| Own irrigation | 0.389 *** | 0.383 *** | |
| Labor hours | 0.000476 *** | 0.000457 *** | 0.000485 *** |
| Improved variety | 0.840 *** | 0.861 *** | 0.845 *** |
| Having livestock | 0.11 | 0.0978 | 0.0816 |
| Constant | 2.073 *** | 2.101 *** | 2.132 *** |
| Participation in extension services | | | |
| Household head's age | 0.00779 *** | 0.00760 *** | 0.00809 *** |

**Table A1.** *Cont.*

| Variables | Ext. Receiver vs. No Ext. | More than One vs. No Ext. | Govt. Ext. vs. No Ext. |
|---|---|---|---|
| Household head's gender | 1.425 *** | 1.420 *** | 1.272 *** |
| Household head's education | 0.0503 *** | 0.0661 *** | 0.0469 *** |
| Agriculture as the main occupation | −0.226 ** | −0.230 ** | −0.188 * |
| Rice area | 0.00143 ** | 0.00167 ** | 0.00142 ** |
| Access to credit | 0.147 *** | 0.154 *** | 0.0823 * |
| Own irrigation | 0.274 *** | 0.325 *** | |
| Labor hours | 0.000170 *** | 0.000149 *** | 0.000157 *** |
| Improved variety | −0.138 *** | −0.0359 | −0.296 *** |
| Having livestock | −0.0999 | −0.141 * | −0.136 * |
| Mobile phone | 0.0958 *** | −0.00782 | 0.0838 *** |
| Distance to market | 0.421 *** | 0.295 ** | 0.400 ** |
| Constant | −3.492 *** | −3.577 *** | −3.235 *** |
| Rho 1 | 0.52 ** | −0.97 *** | 0.34 |
| Rho 2 | 0.69 *** | 0.59 *** | 0.78 *** |
| Observations | 6564 | 6315 | 6440 |
| Prob > chi2 | 0.00 | 0.00 | 0.00 |

\* Significant at the 10% level, ** Significant at the 5% level, *** Significant at the 1% level.

**Table A2.** ESR (Impact on Profit).

| Variables | Ext. Receiver vs. No Ext. | More than One vs. No Ext. | Govt. Ext. vs. No Ext. | Private Ext. vs. No Ext. |
|---|---|---|---|---|
| Per ha net profit (Participation = 1) | | | | |
| Household head's age | −108.7 | 82.84 | −87.84 | −2903 *** |
| Household head's gender | −48,188 *** | | −50,558 ** | |
| Household head's education | −227.6 | −567.2 | −311.9 | |
| Agriculture as the main occupation | 8387 | 14,309 ** | 10,888 | 10,382 |
| Rice area | −47.99 | −71.16 * | 3.217 | −320.5 *** |
| Access to credit | 1680 | −3976 | 4603 * | −13,039 *** |
| Own irrigation | −636.8 | −6001 | −4038 | 16,113 *** |
| Labor hours | −10.53 *** | −9.094 *** | −9.435 *** | −18.41 *** |
| Improved variety | 9030 *** | 10,250 *** | 15,122 *** | −25,028 ** |
| Having livestock | 1900 | 4331 | 3907 | −283.0 * |
| Constant | 70,032 *** | 21,061 | 57,647 * | 215,016 *** |
| Per ha net profit (Participation = 0) | | −3806 | −4023 | −167.2 |
| Household head's age | −230.0 *** | −198.6 *** | −230.1 *** | 130.6 |
| Household head's gender | 15,718 *** | | 16,250 *** | |
| Household head's education | 187.3 * | 181.2 | 221.9 ** | |
| Agriculture as the main occupation | 10,203 *** | 11,090 *** | 9819 *** | 11,100 *** |
| Rice area | −22.67 * | −16.7 | −28.27 ** | −18.19 |
| Access to credit | −2242 *** | −1694 ** | −2177 *** | −2276 *** |
| Own irrigation | 949.5 | 977.6 | 1442 | 2515 ** |

**Table A2.** *Cont.*

| Variables | Ext. Receiver vs. No Ext. | More than One vs. No Ext. | Govt. Ext. vs. No Ext. | Private Ext. vs. No Ext. |
|---|---|---|---|---|
| Labor hours | −4.742 *** | −4.646 *** | −4.584 *** | −4.256 *** |
| Improved variety | 8355 *** | 8972 *** | 8089 *** | 2833 * |
| Having livestock | 1991 | 2098 | 1523 | −243.6 *** |
| Constant | 12,885 *** | 25,003 *** | 13,337 *** | 34,546 *** |
| Participation in extension services | | | | |
| Household head's age | 0.00745 *** | 0.00864 *** | 0.00744 *** | 0.0478 *** |
| Household head's gender | 1.402 *** | | 1.236 *** | |
| Household head's education | 0.0486 *** | 0.0643 *** | 0.0428 *** | |
| Agriculture as the main occupation | −0.256 ** | −0.202 * | −0.214 * | −0.279 |
| Rice area | 0.00132 ** | 0.00156 ** | 0.00104 | 0.000326 |
| Access to credit | 0.173 *** | 0.203 *** | 0.102 ** | 0.264 *** |
| Own irrigation | 0.324 *** | 0.365 *** | 0.406 *** | −0.081 |
| Labor hours | 0.000123 *** | 0.000106 *** | $9.35 \times 10^{-5}$ *** | 0.000144 *** |
| Improved variety | −0.0790 ** | 0.0124 | −0.202 *** | 0.316 ** |
| Having livestock | −0.0749 | −0.0974 | −0.134 * | 0.00346 |
| Mobile phone | 0.0896 *** | 0.0229 | 0.0951 *** | 0.0666 ** |
| Distance to market | 0.540 *** | 0.391 ** | 0.501 *** | 0.0618 |
| Constant | −3.615 *** | −2.430 *** | −3.345 *** | −2.821 *** |
| Rho 1 | 0.12 | 0.03 | 0.14 | −1.87 *** |
| Rho 2 | −0.04 | −0.03 | −0.01 | −0.02 |
| Observations | 6564 | 6315 | 6440 | 6564 |
| Prob > chi2 | 0.00 | 0.00 | 0.00 | 0.00 |

* Significant at the 10% level, ** Significant at the 5% level, *** Significant at the 1% level.

## Appendix B

*Instrumental Validity Test*

**Table A3.** Distance to Bazar.

| Parameter Estimate | Got Ext. Service or Not | Per ha Yield | Per ha Urea Use |
|---|---|---|---|
| Distance to bazar | 0.066 ** (0.02) | −0.04 (0.03) | −2.54 (2.39) |
| Constant | −0.98 *** (0.02) | 3.52 *** (0.02) | 173.72 *** (2.01) |
| Wild test | Chi2 = 5.9 | F = 1.58 | F = 1.13 |
| Observations | 6965 | 6965 | 6965 |

** Significant at the 5% level, *** Significant at the 1% level.

**Table A4.** Use of Mobile Phone.

| Parameter Estimate | Got Ext. Service or Not | Per ha Yield | Per ha Net Profit | Per ha Urea Use |
|---|---|---|---|---|
| Mobile phone use | 0.59 *** (0.15) | 0.11 (0.13) | −1622 (2197) | −7.48 (9.85) |
| Constant | −1.5 *** (0.14) | 3.39 *** (0.13) | 23,065 *** (2169) | 179.9 *** (9.73) |
| Wild test | Chi2 = 18.22 *** | F = 0.76 | F = 0.54 | F = 0.17 |
| Observations | 6599 | 6599 | 6599 | 6599 |

*** Significant at the 1% level.

## Appendix C

*Appendix C.1 Propensity Score Matching*

PSM is widely used non-experimental technique for estimating causal relationships [63]. This statistical technique constructs an artificial comparison group based on a probability

model. It attempts to estimate the probability of every non-treatment and treatment observation receiving treatment based on observed characteristics. In a first, a non-treatment group is identified that is similar to the treatment group in observable chrematistics, and then the impact of treatment is compared for these two groups to identify the treatment effect with correction for self-selection.

PSM involves estimating the propensity score (PS) for each observation. This is the probability of a farmer receiving extension services (P). It is estimated as a probit model with a dependent variable equal to 1 for farmers that received an extension and including a large set of characteristics (X), which can explain the probability of any farmer in the population receiving extension services.

$$PS = Prob\ (P = 1 \mid X) \tag{A1}$$

$$= Prob(\beta_0 + \beta_i X_i + \varepsilon > 0 \tag{A2}$$

The variable vector *X* includes a binary gender variable and the education level of the farmer (years of school completed). A binary agriculture is the main occupation indicator *t*; a binary indicator takes a value of one if the farmer has any livestock holdings or if the farmer used an improved variety of seeds. $\beta_i$ is the vector of the estimated marginal impact of the characteristics on the probability of receiving extension services.

Secondly, we estimate the average treatment effects (ATEs) of receiving extension services on the dependent variables, fertilizer application rate, yield, and net crop income. From the above-mentioned probit regression, we can obtain the PS of the treated group, i.e., those receiving extension services, and the non-treated group. Then we are able to estimate the average difference in welfare between treated Y(1) and matched controlled Y(0) [64–66].

$$ATE = E[Y(1) - Y(0)] = E[Y(1)] - E[Y(0)] \tag{A3}$$

The estimated ATEs are represented as the impact of the extension service on the welfare of the farmers.

However, the treatment effect estimates from the PSM can still be biased in the presence of the misspecification in the propensity score models primarily because of the unobserved attributes of farmers with higher treatment propensity [67–69].

*Appendix C.2 Fertilizer Use*

The impact of extension advice on fertilizer use is shown below:

**Table A5.** PSM estimation of extension service and fertilizer use.

| Name of Fertilizer | Fertilizer Use (PSM) | | | |
| --- | --- | --- | --- | --- |
| | Extension Contacts | More than One Contact | Govt. Ext. Service | Private Ext. Service |
| Urea | 10.7 ** | −4.63 | 11.54 | −10.26 |

** Significant at the 5% level.

The results show that the farmers that received extension contact used significantly higher amounts of urea fertilizer. On the other hand, the relationship between the farmers who received private extension services, government extension services, and those who received more than one extension visit in using urea fertilizer is not significant. Here, private and government extensions show similar results but the extension contact and more than one extension contact groups show different results than the ESR method.

*Appendix C.3 Yield and Income*

The impact of extension advice on yield and farm profit is shown below:

**Table A6.** PSM estimation of extension service on rice yield and profit.

| | PSM | | | |
|---|---|---|---|---|
| | Extension Contacts | More than One Contact | Govt. Ext. Service | Private Ext. Service |
| Per hectare yield | 0.33 ** | −0.05 | 0.25 ** | 0.52 ** |
| Field level profit | 3432 ** | 2366 | 2349 | 8543 *** |

** Significant at the 5% level, *** Significant at the 1% level.

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
