# Peer review of "Impact of Agricultural Extension Services on Fertilizer Use and Farmers’ Welfare: Evidence from Bangladesh"

_sustainability, doi:10.3390/su14159385_

Round 1

Reviewer 1 Report

  1. The abstract lacks the spatial and time scope of the research.
  2. Keywords – impact??
  3. I would suggest improving the title of section 2.
  4. The methodology indicates the database used in the research - Bangladesh Integrated Household Survey-BIHS (2015). This description shows that the 2015 questionnaires were taken into account. The question arises as to how the use of these data can be justified from the point of view of 2022. This requires some explanation.
  5. In Table 1, the measurement unit was not given for some variables.
  6. I suggest making an editorial correction, including removing the yellow highlighting of the text.

Reviewer 2 Report

Dear authors,

A manuscript has been accepted with minor revisions,

my minor corrections in attached revised copy

Best regards,

Reviewer 3 Report

Thanks for choosing me (as the referee) to review the manuscript entitled “Impact of Agricultural Extension service on Fertilizer use and Farmer’s Welfare: Evidence from Bangladesh (Manuscript ID:  sustainability-1701153)”, the following points will be provided:

The paper falls within the aims and scopes of journal. Also, the paper is reasonably well written and it reads well, the study in itself seems interesting enough for its report to be upgraded and further prepared for international publication. For this purpose, to be achieved, however, several things are needed, as follows:

  • Title: Good title.
  • Abstract: The authors need to add the summary of research methodology. It is missing in the abstract. What are the policy recommendations from this study? What is the "effect" of the research as a general conclusion? This effect needs to be added at the end of the abstract.
  • Keywords: Avoid repeating the words in the title.
  • Introduction: Many of the references are old.
  • Context: What is the ontology and epistemology (theoretical context) governing this research?
  • Methodology: In general, what was the kind of research method in this study (Paradigm, type, data, time, gathering data, …)
  • Results: Well, presented.
  • Discussion and Conclusion: This part needs further strengthening. What are the policy recommendations? What are the suggestions for future research? What is the overall effect of the paper? What were the limitations of the research?

With regard to what was said, my suggestion is to accept the manuscript after minor revisions”.
